# Robustness Guarantees for Adversarially Trained Neural Networks

**Poorya Mianjy**
Johns Hopkins University

**Raman Arora**
Johns Hopkins University
arora@cs.jhu.edu

## Abstract

We study robust adversarial training of two-layer neural networks as a bi-level optimization problem. In particular, for the inner loop that implements the adversarial attack during training using projected gradient descent (PGD), we propose maximizing a *lower bound* on the $0/1$-loss by reflecting a surrogate loss about the origin. This allows us to give a convergence guarantee for the inner-loop PGD attack. Furthermore, assuming the data is linearly separable, we provide precise iteration complexity results for end-to-end adversarial training, which holds for any width and initialization. We provide empirical evidence to support our theoretical results.

## 1 Introduction

Despite the tremendous success of deep learning, neural network-based models are highly susceptible to small, imperceptible, adversarial perturbations of data at test time [Szegedy et al., 2014]. Such vulnerability to *adversarial examples* imposes severe limitations on the deployment of neural networks-based systems, especially in critical high-stakes applications such as autonomous driving, where safe and reliable operation is paramount.

An abundance of studies demonstrating adversarial examples across different tasks and application domains [Goodfellow et al., 2014, Moosavi-Dezfooli et al., 2016, Carlini and Wagner, 2017] has led to a renewed focus on robust learning as an active area of research within machine learning. The goal of *robust learning* is to find models that yield reliable predictions on test data notwithstanding adversarial perturbations. A principled approach to training models that are robust to adversarial examples that has emerged in recent years is that of *adversarial training* [Madry et al., 2018]. Adversarial training formulates learning as a min-max optimization problem wherein the 0-1 classification loss is replaced by a convex surrogate such as the cross-entropy loss, and alternating optimization techniques are used to solve the resulting saddle point problem.

Despite empirical success of adversarial training, our understanding of its theoretical underpinnings remain limited. From a practical standpoint, it is remarkable that gradient based techniques can efficiently solve both inner maximization problem to find adversarial examples and outer minimization problem to impart robust generalization. On the other hand, a theoretical analysis is challenging because (1) both the inner- and outer-level optimization problems are non-convex, and (2) it is unclear a-priori if solving the min-max optimization problem would even guarantee robust generalization.

In this work, we seek to understand adversarial training better. In particular, under a margin separability assumption, we provide robust generalization guarantees for two-layer neural networks with Leaky ReLU activation trained using adversarial training. Our key contributions are as follows.

1. We identify a disconnect between the robust learning objective and the min-max formulation of adversarial training. This observation inspires a simple modification of adversarial training –

37th Conference on Neural Information Processing Systems (NeurIPS 2023).

we propose *reflecting* the surrogate loss about the origin in the inner maximization phase when searching for an "optimal" perturbation vector to attack the current model.

2. We provide convergence guarantees for PGD attacks on two-layer neural networks with leaky ReLU activation. This is the first of its kind result to the best of our knowledge.

3. We give global convergence guarantees and establish learning rates for adversarial training for two-layer neural networks with Leaky ReLU activation function. Notably, our guarantees hold for *any bounded initialization* and *any width* – a property that is not present in the previous works in the neural tangent kernel (NTK) regime [Gao et al., 2019, Zhang et al., 2020].

4. We provide extensive empirical evidence showing that reflecting the surrogate loss in the inner loop does not have a significant impact on the test time performance of the adversarially trained models.

**Notation.** We denote matrices, vectors, scalar variables, and sets by Roman capital letters, Roman lowercase letters, lowercase letters, and uppercase script letters, respectively (e.g. X, x, $x$, and $\mathcal{X}$). For any integer $d$, we represent the set $\{1, \ldots, d\}$ by $[d]$. The $\ell_2$-norm of a vector x and the Frobenius norm of a matrix X are denoted as $\|x\|$ and $\|X\|_F$, respectively. Given a set $\mathcal{C}$, the operator $\Pi_{\mathcal{C}}(x) = \min_{x' \in \mathcal{C}} \|x - x'\|$ projects onto the set $\mathcal{C}$ with respect to the $\ell_2$-norm.

## 1.1 Related Work

**Linear models.** Adversarial training of linear models was recently studied by Charles et al. [2019], Li et al. [2020], Zou et al. [2021]. In particular, Charles et al. [2019], Li et al. [2020] give robust generalization error guarantees for adversarially trained linear models under a margin separability assumption. The hard margin assumption was relaxed by Zou et al. [2021] who give robust generalization guarantees for distributions with agnostic label noise. We note that the optimal attack for linear models has a simple closed-form expression, which mitigates the challenge of analyzing the inner loop PGD attack. In contrast, one of our main contributions is to give convergence guarantees for the PGD attack. Nonetheless, as the Leaky ReLU activation function can also realize the identity map for $\alpha = 1$, our results also provide robust generalization error guarantees for training linear models.

**Non-linear models.** Wang et al. [2019] propose a first order stationary condition to evaluate the convergence quality of adversarial attacks found in the inner loop. Zhang et al. [2021] study adversarial training as a bi-level optimization problem and propose a principled approach towards the design of fast adversarial training algorithms. Most related to our results are the works of Gao et al. [2019] and Zhang et al. [2020], which study the convergence of adversarial training in non-linear neural networks. Under specific initialization and width requirements, these works guarantee small robust training error with respect to the attack that is used in the inner-loop, without explicitly analyzing the convergence of the attack. Gao et al. [2019] assume that the activation function is smooth and require that the width of the network, as well as the overall computational cost, is exponential in the input dimension. The work of Zhang et al. [2020] partially addresses these issues. In particular, their results hold for ReLU neural networks, and they only require the width and the computational cost to be polynomial in the input parameters.

Our work is different from that of Gao et al. [2019] and Zhang et al. [2020] in several ways. Here we highlight three key differences.

- First, while the prior work analyzes the convergence in the NTK setting with specific initialization and width requirements, our results hold for any initialization and width.

- Second, none of the prior works studies computational aspects of finding an optimal attack vector in the inner loop. Instead, the prior work assumes oracle access to optimal attack vectors. We provide precise iteration complexity results for the projected gradient method (i.e., for the PGD attack) for finding near-optimal attack vectors.

- Third, the prior works focus on minimizing the robust training loss, whereas we provide computational learning guarantees on the robust generalization error.

The rest of the paper is organized as follows. In Section 2, we give the problem setup and introduce the adversarial training procedure with the reflected surrogate loss in the inner loop. In Section 3, we present our main results, discuss the implications and give a proof sketch. We support our theory with empirical results in Section 4 and conclude with a discussion in Section 5.

**Algorithm 1** `Atk` PGD Attack
***
**Input:** Sample $(\mathrm{x}, y)$, Weights W, Stepsize $\eta_{\mathtt{atk}}$, # Iters $T_{\mathtt{atk}}$
1: Initialize $\delta_1 \leftarrow \mathrm{x}$
2: **for** $t = 1$ to $T$ **do**
3:     $\delta_{t+1} \leftarrow \Pi_{\Delta(\mathrm{x})}(\delta_t + \eta_{\mathtt{atk}} \nabla_\delta \ell_-(y f_{\mathrm{W}}(\delta_t)))$
4: **end for**
**Output:** $\delta_\tau$, where $\tau \in \arg\max_{t \in [T]} \ell_-(y f_{\mathrm{W}}(\delta_t))$
***

## 2 Preliminaries

We focus on two-layer networks with $m$ hidden nodes computing $f(\mathrm{x}; \mathrm{a}, \mathrm{W}) = \mathrm{a}^\top \sigma(\mathrm{Wx})$, where $\mathrm{W} \in \mathbb{R}^{m \times d}$ and $\mathrm{a} \in \mathbb{R}^m$ are the weights of the first and the second layers, respectively, and $\sigma(z) = \max\{\alpha z, z\}$ is the Leaky ReLU activation function. We randomly initialize the weights $a_r \sim \mathrm{Unif}(\{-\kappa, +\kappa\})$ for all hidden nodes $r \in [m]$, and randomly initialize W such that $\|\mathrm{W}\|_F \le \omega$, for some positive real numbers $\kappa$ and $\omega$. The top linear layer (i.e., weights a) is kept fixed, and the hidden layer (i.e., W) is trained using stochastic gradient descent (SGD).

For simplicity of notation, we represent the network as $f(\mathrm{x}; \mathrm{W})$, suppressing the dependence on the top layer weights. Further, with a slight abuse of notation, we denote the function by $f_{\mathrm{W}}(\mathrm{x})$ when optimizing over the input adversarial perturbations, and by $f_{\mathrm{x}}(\mathrm{W})$ when training the network weights.

Formally, adversarial learning is described as follows. Let $\mathcal{X} \subseteq \mathbb{R}^d$ and $\mathcal{Y} = \{\pm 1\}$ denote the input feature space and the output label space, respectively. Let $\mathcal{D}$ be an unknown joint distribution on $\mathcal{X} \times \mathcal{Y}$. For any fixed $\mathrm{x} \in \mathcal{X}$, we consider norm-bounded adversarial perturbations in the set $\Delta(\mathrm{x}) := \{\delta : \|\delta - \mathrm{x}\| \le \nu\}$, for some fixed noise budget $\nu$.

Given a training sample $\mathcal{S} := \{(\mathrm{x}_i, y_i)\}_{i=1}^n \sim \mathcal{D}^n$ drawn independently and identically from the underlying distribution $\mathcal{D}$, the goal is to find a network with small robust misclassification error

$$\varepsilon_{\mathrm{rob}}(\mathrm{W}) = \mathbb{E}_{\mathcal{D}} \max_{\delta \in \Delta(\mathrm{x})} \mathbb{I}[y f_{\bar{\mathrm{W}}}(\delta) < 0], \tag{1}$$

where $\bar{\mathrm{W}} := \mathrm{W}/\|\mathrm{W}\|_F$ is the weight matrix normalized to have unit Frobenius norm. Note that, due to the homogeneity of Leaky ReLU, such normalization has no effects on the robust error whatsoever.

In adversarial training, the $0 - 1$ loss inside the expectation is replaced with a convex surrogate such as cross entropy loss $\ell(z) = \log(1 + e^{-z})$, and the expected value is estimated using a sample average:

$$\widehat{\varepsilon}_{\mathrm{rob}}(\mathrm{W}) := \frac{1}{n} \sum_{i=1}^n \max_{\delta_i \in \Delta(\mathrm{x}_i)} \ell(y_i f_{\bar{\mathrm{W}}}(\delta_i)) \tag{2}$$

Notwithstanding the conventional wisdom, adversarial training entails *maximizing* an *upper bound* as opposed to a *lower bound* on the $0 - 1$ loss. In contrast, we propose using a *concave lowerbound* on the $0 - 1$ loss to solve the inner maximization problem. Let $\ell_-(z) = -\ell(-z) = -\log(1 + e^z)$ denote the *reflected loss*. In Figure 1, we plot the 0-1 loss, the cross-entropy loss, and the reflected cross-entropy loss. Starting from $\delta_1 = \mathrm{x}$, the PGD attack updates iterates via

$$\delta_{t+1} = \Pi_{\Delta(\mathrm{x})}(\delta_t + \eta_{\mathtt{atk}} \ell_-(y f_{\mathrm{W}}(\delta_t))),$$

as described in Algorithm 1. We emphasize that the only difference between standard adversarial training and what we propose in Algorithm 2 and Algorithm 1 is that we reflect the loss (about the origin) in Algorithm 1.

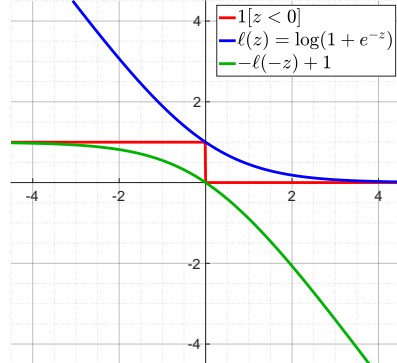

Figure 1: The 0-1 loss (red), its convex surrogate, the cross-entropy loss (blue), and the reflected cross-entropy loss (green).

**Algorithm 2** `AdvTr` Adversarial Training

---

**Input:** Stepsize $\eta_{\tt tr}$, # Iters $T_{\tt tr}$
1: Initialize $a \sim \mathrm{Unif}\{-\kappa, +\kappa\}^m$ and $W_1$ such that $\|W_1\|_F \leq \omega$
2: **for** $t = 1$ to $T$ **do**
3:     Draw $(x_t, y_t) \sim \mathcal{D}$
4:     $\delta_t \leftarrow \mathtt{Atk}(W_t, x_t, y_t)$
5:     $W_{t+1} \leftarrow W_t - \eta_{\tt tr}\nabla_W \ell(y_t f_{\delta_t}(W_t))$
6: **end for**

---

## 3  Main Results

We consider a slightly weaker version of the robust error. In particular, we are interested in adversarial attacks that can fool the learner with a margin – for some small, non-negative constant $\beta$, we define the $\beta$-*robust misclassification error* as: $\varepsilon_\beta(W) = \mathbb{P}\left\{\min_{\delta \in \Delta(x)} y f_{\bar{W}}(\delta) < -\beta\right\}$. In particular, as $\beta$ tends to zero, $\varepsilon_\beta(W) \to \varepsilon_{\mathrm{rob}}(W)$. When $\beta$ is a small positive constant bounded away from zero, $(x, y)$ contributes to $\varepsilon_\beta(W)$ only if there exists an *attack* $\delta \in \Delta(x)$ such that $f_{\bar{W}}$ *confidently* makes a wrong prediction on $\delta$. In other words, $\beta$-robust misclassification error is the probability that for $(x, y) \sim \mathcal{D}$, a $\beta$-*effective attack* exists.

**Definition 3.1** (Effective Attacks). Given a neural networks with parameters $(a, W)$ and a data point $(x, y)$ and some constant $\beta > 0$, we say that $\delta_* \in \Delta(x)$ is a $\beta$-effective attack if $y f_{\bar{W}}(\delta_*) \leq -\beta$, where $\bar{W} = W/\|W\|_F$.

Our bounds depend on several important problem parameters. Before stating the main results of the paper, we remind the reader of these important quantities. $\nu$ denotes the attack size. $\kappa$ and $\omega$ are the bounds on the norm of the parameters $a$ and $W$ at the initialization. Finally, $\alpha$ is the Leaky-ReLU parameter. Our first result stated in the following theorem[1] gives convergence rates for Algorithm 1 in terms of the and the negated loss derivative $-\ell'(\cdot)$ under the assumption that an effective attack exists. The negative derivative, $-\ell'(\cdot)$, of the loss function has been used in several previous works to give an upper bound on the error Cao and Gu [2019]; here, we borrow similar ideas from Frei et al. [2021]. In particular, as it will become clear later, we will use positivity and monotonicity of $-\ell'(\cdot)$ to give an upper bound on the $\beta$-robust loss using Markov's inequality.

**Theorem 3.2.** *Let $\delta_*$ be a $\beta$-effective attack for a given network with weights $(a, W)$ and a given example $(x, y)$, with $\beta \geq 2\nu(1 - \alpha)\kappa\sqrt{m}$. Then, after $T_{\tt atk}$ iterations, PGD with step size $\eta_{\tt atk} \leq \frac{1}{\kappa^2 m \|W\|_F^2}$ generates an attack $\delta_{\tt atk}$ such that $-\ell'(y f_W(\delta_*)) \leq -2\ell'(y f_W(\delta_{\tt atk})) + \frac{4\nu^2}{\eta_{\tt atk} T_{\tt atk}}$.*

Theorem 3.2 establishes that under proper initialization ($\kappa = 1/\sqrt{m}$), when a $\beta$-effective attack exists, Algorithm 1 finds a $\epsilon$-suboptimal attack vector in $O(\beta^2/\epsilon)$ iteration. We next study convergence of Algorithm 2 under the following distributional assumption.

**Assumption 3.3.** Samples $(x, y)$ are drawn i.i.d. from an unknown joint distribution $\mathcal{D}$ that satisfies:

- $\|x\| \leq R$ with probability 1.

- There exists a unit norm vector $v_* \in \mathbb{R}^d$, $\|v_*\| = 1$, such that for $(x, y) \sim \mathcal{D}$, we have with probability 1 that $y(v_* \cdot x) \geq \gamma > 0$.

The first assumption requires that the inputs are bounded, which is standard in the literature and is satisfied for most practical applications. The second assumption implies that $\mathcal{D}$ is linearly separable with margin $\gamma > 0$. Of course, we do not need a non-linear neural network to robustly learn a predictor under such a distributional assumption. But can we even guarantee robust learnability of neural networks for such simple settings? Nothing is known as far as we know. We note that even for standard (non-robust) training of two-layer neural networks using SGD, the convergence guarantees in the hard margin setting were unknown until recently [Brutzkus et al., 2018]. The following theorem establishes that adversarial training can efficiently find a network with small $\beta$-robust error.

---

[1]Proofs are deferred to the appendix.

**Theorem 3.4** (Convergence of Algorithm 2). *For any $\epsilon > 0$, in at most $T_{tr} \leq \frac{64(R+\nu)^2(1+\omega\gamma\alpha\kappa\sqrt{m}\epsilon)}{(\gamma-\nu)^2\alpha^2\epsilon^2}$ iterations, Algorithm 2 with step-size $\eta_{tr} \leq \frac{1}{m\kappa^2(R+\nu)^2}$ finds an iterate $\tau$ that, in expectation over $\{(x_t, y_t)\}_{t=1}^{T_{tr}}$, satisfies $\varepsilon_\beta(W_\tau) \leq 2\epsilon$ for any $\beta \geq 2\nu(1-\alpha)\kappa\sqrt{m}$, provided that for all $t \in [T]$, $\eta_{atk} \leq \frac{1}{\kappa^2 m\|W_t\|_F^2}$ and $T_{atk} \geq \frac{8\nu^2}{\eta_{atk}\epsilon}$.*

Some remarks are in order.

**Beyond Neural Tangent Kernel.** As opposed to the convergence results in the previous work [Gao et al., 2019, Zhang et al., 2020] which requires certain initialization and width requirements specific to the NTK regime, our results holds for *any bounded initialization* and *any width $m$*.

**Role of the Robustness Parameter $\nu$.** Our guarantee holds only when the desired robustness parameter $\nu$ is smaller than the distribution margin $\gamma$. Furthermore, the iteration complexity increases gracefully as $O(\nu^2/(\gamma - \nu)^2)$ as the *attacks* become stronger, i.e., as the size of adversarial perturbations tends to the margin. Intuitively, as $\nu \to 0$, the attack becomes trivial, and the adversarial training reduces to the standard non-adversarial training. This is fully captured by our results — as $\nu \to 0$, the number of attack iterates $T_{atk}$ goes to zero, and we recover the overall runtime of $O(\gamma^{-2}\epsilon^{-2})$ as in the previous work [Brutzkus et al., 2018, Frei et al., 2021].

**Computational Complexity.** To guarantee $\epsilon$-suboptimality in the $\beta$-robust misclassification error, we require $T_{tr} = O((\gamma - \nu)^{-2}\epsilon^{-2})$ iterations of Algorithm 2. Each iteration invokes the PGD attack in Algorithm 1, which itself requires $T_{atk} = O(\nu^2/\epsilon)$ gradient updates. Therefore, the overall computational cost of adversarial training to achieve $\epsilon$-suboptimality is $O(\frac{\nu^2}{(\gamma-\nu)^2\epsilon^3})$. Note that $T_{atk}$ is a purely computational requirement, and the statistical complexity of adversarial training is fully captured by $T_{tr}$. Remarkably, there is only a mild $O(\gamma^2/(\gamma - \nu)^2)$ statistical overhead for $\beta$-robustness, and the computational cost increases gracefully by a multiplicative factor of $O\left(\frac{\nu^2\gamma^2}{(\gamma-\nu)^2\epsilon}\right)$.

**Learning Robust Linear Halfspaces.** When $\alpha = 1$, the Leaky ReLU activation equals the identity map, and the network reduces to a linear predictor. In this case, we retrieve strong robust generalization guarantees for learning halfspaces, as the lower bound required for $\beta$ in Theorem 3.4 vanishes. The following corollary instantiates such a robust generalization guarantee.

**Corollary 3.5.** *Let $\kappa = 1/\sqrt{m}$, $\omega = 1/\gamma$, and $\eta_{tr} = (R + \nu)^{-2}$. For any $\epsilon > 0$, in at most $T_{tr} \leq \frac{128(R+\nu)^2}{(\gamma-\nu)^2\epsilon^2}$ iterations, Algorithm 2 finds an iterate $\tau$, that in expectation over $\{(x_t, y_t)\}_{t=1}^{T_{tr}}$, satisfies $\varepsilon_{rob}(W_\tau) \leq 2\epsilon$, provided that for all $t \in [T]$, $\eta_{atk} \leq \|W_t\|_F^{-2}$ and $T_{atk} \geq \frac{8\nu^2}{\eta_{atk}\epsilon}$.*

**Dependence on the Norm of Iterates.** The iteration complexity of Algorithm 1 is inversely proportional to the learning rate $\eta_{atk}$, and therefore increases with $\|W_t\|_F^2$. Thus, when calculating the overall computational complexity, one needs to compute an upper bound on the norm of the iterates. As we show in Equation (5) in the appendix, it holds for all iterates that $\|W_{t+1}\|_F^2 \leq \|W_1\|_F^2 + 3\eta_{tr}t$. Therefore, if we set $\kappa = 1/\sqrt{m}$ and $\omega^2 = 3/(R + \nu)^2$, we have the following worst-case weight-independent bound on the overall computational cost:

$$T \leq \sum_{t=1}^{T_{tr}} \frac{8\nu^2}{\eta_{atk}\epsilon} \leq \sum_{t=1}^{T_{tr}} \frac{8\nu^2\|W_t\|_F^2}{\epsilon} \leq \sum_{t=1}^{T_{tr}} \frac{8\nu^2(\omega^2 + 3\eta_{tr}(t-1))}{\epsilon}$$

$$\leq \sum_{t=1}^{T_{tr}} \frac{24\nu^2 t}{(R+\nu)^2\epsilon} \leq \frac{12\nu^2 T_{tr}^2}{(R+\nu)^2\epsilon} \leq \frac{196608\nu^2(R+\nu)^2}{(\gamma-\nu)^4\alpha^4\epsilon^5}.$$

Therefore, the worst-case overall computational cost is of order $O((\gamma - \nu)^{-4}\epsilon^{-5})$. We note again that this cost is purely computational – the statistical complexity is still in the order of $O\left((\gamma - \nu)^{-2}\epsilon^{-2}\right)$.

**Adversarial Robustness for any $\beta$.** As we discussed earlier, as $\beta \to 0$, the $\beta$-robust error tends to the robust error, i.e., $\varepsilon_\beta(W) \to \varepsilon_{rob}(W)$. Although Theorem 3.4 does not hold for $\beta = 0$ (except

for the linear case discussed above), it is possible to guarantee robust generalization with arbitrarily small $\beta$, as stated in the following corollary.

**Corollary 3.6.** *For any desirable $\beta > 0$, let $\kappa = \frac{\beta}{2\nu(1-\alpha)\sqrt{m}}$. For any $\epsilon > 0$, in at most $T_{tr} \leq \frac{64(R+\nu)^2(1+\omega\gamma\alpha\beta\epsilon/(2\nu(1-\alpha)))}{(\gamma-\nu)^2\alpha^2\epsilon^2}$ iterations, Algorithm 2 with step-size $\eta_{tr} \leq \frac{4\nu^2(1-\alpha)^2}{\beta^2(R+\nu)^2}$ finds an iterate $\tau$ that, in expectation over $\{(x_t, y_t)\}_{t=1}^{T_{tr}}$, satisfies $\varepsilon_\beta(W_\tau) \leq 2\epsilon$ provided that for all $t \in [T]$, $\eta_{atk} \leq \frac{4\nu^2(1-\alpha)^2}{\beta^2\|W_t\|_F^2}$ and $T_{atk} \geq \frac{2(1-\alpha)^2}{\beta^2\|W_t\|_F^2\epsilon}$.*

## 3.1 Proof Sketch

In this section, we highlight the key ideas and insights based on our analysis, and give a sketch of the proof of the main result. Using Definition 3.1, the proof of Theorem 3.4 crucially depends on the following two facts. First, whenever there exists a $\beta$-effective attack, we show that Algorithm 1 will efficiently find a sufficiently good attack (in the sense of Theorem 3.2). Second, leveraging the Perceptron analysis from the prior works of Frei et al. [2021], Brutzkus et al. [2018], we show that as long as the attack size $\nu$ is smaller than the margin $\gamma$, robust training is not much harder than standard training. In particular, the following Lemma establishes that the expected value of the negative loss derivative eventually becomes arbitrarily small.

**Lemma 3.7.** *For any $\epsilon > 0$, Algorithm 2 with stepsize $\eta_{tr} \leq m^{-1}\kappa^{-2}(R+\nu)^{-2}$ finds an iterate $\tau$ that, in expectation over $\{(x_t, y_t)\}_{t=1}^{T_{tr}}$, satisfies $\mathbb{E}_{\mathcal{D}}[-\ell'(yf_{W_\tau}(\delta_{atk}(x)))] \leq \epsilon$ in at most $T_{tr} \leq \frac{4(1+\|W_1\|_F\gamma\alpha\kappa\sqrt{m}\epsilon)}{\eta_{tr}(\gamma-\nu)^2\alpha^2\kappa^2 m\epsilon^2}$ iterations.*

We remark that the result in Lemma 3.7 holds for *any* attack algorithm Atk, as long as it respects the condition $\delta_{atk}(x) \in \Delta(x)$ for all x. We are now ready to present the proof of the main result.

*Proof of Theorem 3.4.* Recall, that $\beta$-robust misclassification error is defined as:

$$\varepsilon_\beta(W) = \mathbb{P}\left\{\min_{\delta\in\Delta(x)} yf_{\bar{W}}(\delta) < -\beta\right\} = \mathbb{P}\left\{\min_{\delta\in\Delta(x)} yf_W(\delta) < -\beta\|W\|_F\right\} \quad \text{(Homogeneity of } f\text{)}$$

A key step in the proof is to give an upper bound on $\epsilon_\beta$ in terms of the attack returned by PGD, i.e., $\delta_{atk(x)}$, rather than the optimal attack $\min_{\delta\in\Delta(x)} yf_W(\delta)$. Theorem 3.2 does provide us with such an upper bound; however, (1) it only holds in expectation, and 2) it is conditioned on existence of an effective attack at the given example $(x, y)$ and the weights W. Naturally, we can use Markov's inequality to bound the probability above. In order to address the conditional nature of the result in Theorem 3.2, we introduce a truncated version of the negative loss derivative. In particular, for any $c$, let $\ell'_c(z) = \ell'(z)\mathbb{I}[z \leq c]$ be the loss derivative thresholded at $c$. Note that $z \leq c$ implies that $-\ell'_c(z) \geq -\ell'_c(c)$ – therefore, $\mathbb{P}\{z \leq c\} \leq \mathbb{P}\{-\ell'_c(z) \geq -\ell'_c(c)\}$. Let $\beta_\tau := \beta\|W_\tau\|_F$, where $W_\tau$ is the iterate guaranteed by Lemma 3.7. We have

$$\varepsilon_\beta(W_\tau) = \mathbb{P}\left\{\min_{\delta\in\Delta(x)} yf_{W_\tau}(\delta) \leq -\beta_\tau\right\} \leq \mathbb{P}\left\{-\ell'_{-\beta_\tau}(\min_{\delta\in\Delta(x)} yf_{W_\tau}(\delta)) \geq -\ell'_{-\beta_\tau}(-\beta_\tau)\right\}$$

$$\leq \frac{\mathbb{E}_{\mathcal{D}}\left[-\ell'_{-\beta_\tau}(\min_{\delta\in\Delta(x)} yf_{W_\tau}(\delta))\right]}{-\ell'_{-\beta_\tau}(-\beta_\tau)} \quad \text{(Markov's inequality)}$$

$$\leq 2\mathbb{E}_{\mathcal{D}}\left[-\ell'_{-\beta_\tau}(\min_{\delta\in\Delta(x)} yf_{W_\tau}(\delta))\right] \quad (-\ell'_{-\beta_\tau}(z) \geq 1/2 \text{ for } z \leq 0)$$

Given $W_\tau$, for any $(x, y) \sim \mathcal{D}$, one of the two following cases can happen:

1. *There exists a $\beta$-effective attack.* In this case, by Definition 3.1, it holds that $\min_{\delta\in\Delta(x)} yf_{W_\tau}(\delta) \leq -\beta\|W_\tau\|_F = -\beta_\tau$. Therefore, by definition of the truncated negative loss derivative, it also holds that $-\ell'_{\beta_\tau}(\min_{\delta\in\Delta(x)} yf_{W_\tau}(\delta)) = -\ell'(\min_{\delta\in\Delta(x)} yf_{W_\tau}(\delta))$. Now, using Theorem 3.2, we get that

$$-\ell'_{-\beta_\tau}(\min_{\delta\in\Delta(x)} yf_{W_\tau}(\delta)) \leq -2\ell'(yf_{W_\tau}(\delta_{atk}(x))) + \frac{4\nu^2}{\eta_{atk}T_{atk}}$$

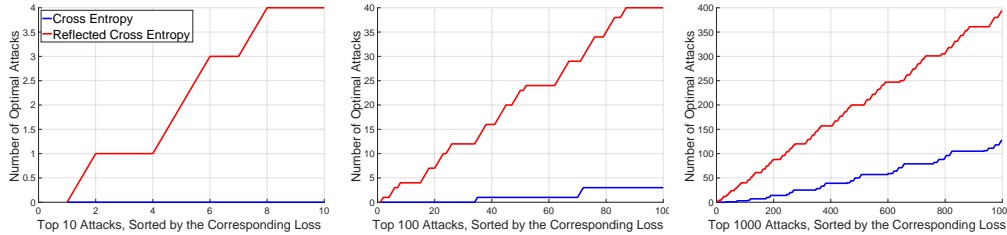

Figure 2: Number of the top-k attack vectors that are optimal, i.e., can induce a label flip, for the cross entropy loss (blue) and the reflected version (red), for different values of $k$: **Left:** $k = 10$, **Middle:** $k = 100$, and **Right:** $k = 1000$.

2. *There does not exist a $\beta$-effective attack.* In this case, by Definition 3.1, it holds that $\min_{\delta \in \Delta(\mathrm{x})} y f_{\mathrm{W}_\tau}(\delta) > -\beta \|\mathrm{W}_\tau\|_F = -\beta_\tau$. Therefore, by definition of the truncated negative loss derivative, it also holds that $-\ell'_{\beta_\tau}(\min_{\delta \in \Delta(\mathrm{x})} y f_{\mathrm{W}_\tau}(\delta)) = 0$, which is trivially bounded by the upper bound in the first case above, given by Equation (3).

Putting back the above cases in the upper bound on the $\beta$-robust error, we arrive at:

$$\frac{1}{2} \varepsilon_\beta(\mathrm{W}_\tau) \le 2 \mathbb{E}_{\mathcal{D}}[-\ell'(y f_{\mathrm{W}_\tau}(\delta_{\mathtt{atk}}(\mathrm{x})))] + \frac{4\nu^2}{\eta_{\mathtt{atk}} T_{\mathtt{atk}}} \le \frac{\epsilon}{2} + \frac{4\nu^2}{\eta_{\mathtt{atk}} T_{\mathtt{atk}}} \le \frac{\epsilon}{2} + \frac{\epsilon}{2}$$

where the first inequality follows from Theorem 3.2, the second inequality follows from Lemma 3.7 given the proper choice of $T_{\mathtt{Tr}}$, and the final inequality holds by setting $T_{\mathtt{atk}} \ge \frac{8\nu^2}{\eta_{\mathtt{atk}} \epsilon}$.  □

## 4 Empirical Results

Adversarial training is widely used in training robust models and has been shown to be fairly effective in practice. The goal of this section is not to attest or reproduce previous empirical findings. Instead, since the focus in this paper is on the theoretical analysis of adversarial training in non-linear networks, the goal of this section is merely to empirically study the effect of using reflected loss in Algorithm 1.

The experimental results are organized as follows. First, in Sec. 4.1, we compare the optimal attacks found by a grid search on the surrogate loss and its reflected version. In Sec. 4.2, we empirically study adversarial training with reflected loss in the binary classification setting. Finally, in Sec. 4.3, we generalize the reflected loss, which is key to our theoretical analysis, to multi-class classification setting. We then report the results on the CIFAR-10 dataset using a deep residual network.

### 4.1 Grid Search Optimization

We look at the following simple 3-dimensional 3-class classification problem. Consider the point $(\mathrm{x}, y)$ where $\mathrm{x} = [3, 2, 1]$ and $y = 1$. We focus on the simplest non-trivial function, i.e., the identity mapping, given by $f(\mathrm{x}) = \mathrm{x}$. Obviously, $f$ correctly assigns $\mathrm{x}$ to the first class because the first dimension is larger than the others. Also, a perturbation of the form $\delta = [-0.501, 0.5, 0]$ with $\|\delta\| = 0.7078$ can flip the label, since $f(\mathrm{x} + \delta) = [2.499, 2.5, 1]$ incorrectly predicts the second class.

We restrict the attack to the set $\{\delta \in (-0.51, +0.51)^3 | \|\delta\| \le 0.7078\}$. We look at every possible attack vector on a grid of size $800 \times 800 \times 800$. We then sort these vectors in a descending order of the corresponding loss function, i.e., the cross entropy loss and its reflected version, and simply count how many of the top-$k$ attack vectors actually induce a label flip. We take this as a measure of how effective is the corresponding loss maximization problem at finding a good attack vector. As we can see in Figure 2, the proposed method of maximizing the reflected cross entropy loss is a far more effective way of generating the attacks than maximizing the cross entropy loss.

### 4.2 Binary Classification

**Experimental Setup.** We extract digits 0 and 1 from the MNIST dataset [LeCun et al., 1998], which provides a (almost) separable distribution, consistent with our theoretical setup. The dataset contains

| Atk.
Trg. | FGSM | R-FGSM | PGD-∞ | R-PGD-∞ | BIM | R-BIM | PGD-2 | R-PGD-2 |
|---|---|---|---|---|---|---|---|---|
| Standard | 0.236 | 0.236 | 0.033 | 0.286 | 0.286 | 0.286 | 0.003 | 0.256 |
| PGD-∞ | 0.004 | 0.004 | 0.005 | 0.005 | 0.005 | 0.005 | 0.003 | 0.05 |
| R-PGD-∞ | 0.003 | 0.003 | 0.004 | 0.004 | 0.004 | 0.004 | 0.002 | 0.042 |
| PGD-2 | 0.013 | 0.013 | 0.022 | 0.024 | 0.024 | 0.024 | 0.002 | 0.034 |
| R-PGD-2 | 0.004 | 0.004 | 0.005 | 0.006 | 0.006 | 0.006 | 0.0 | 0.008 |

Table 1: Robust test error of several adversarially trained models with and without reflecting the loss (Standard training, PGD-∞, R-PGD-∞, PGD-2, R-PGD-2), for different attack benchmarks (FGSM, R-FGSM, PGD-∞, R-PGD-∞, BIM, R-BIM, PGD-2, and R-PGD-2).

12665 training samples and 2115 test samples. We evaluate the generalization error as well as the robust generalization error of fully-connected two-layer neural networks which are adversarially trained with and without reflecting the loss. The network has 100 hidden nodes with ReLU activations.

The outer loop consists of 20 epochs over the training data with batch size equal to 64, randomly shuffled at the beginning of each epoch. The initial learning rate is set to 1, and is decayed by a multiplicative factor of 0.2 every 5 epochs. We use several benchmark attacks with and without reflecting the loss. The benchmarks include the Fast Gradient Sign Method (FGSM) Goodfellow et al. [2015], the Basic Iterative Method (BIM) Kurakin et al. [2017], and the PGD attack with $\ell_2$ constraint (PGD-2) and $\ell_\infty$ constraint (PGD-∞). For each of these attack strategies, we have a corresponding approach that involves reflecting the surrogate loss – we denote the resulting methods as R-FGSM, R-BIM, R-PGD-2, and R-PGD-∞, respectively. The perturbation size for FGSM, PGD-∞, and BIM (and their corresponding reflected version) is set to $\nu = 0.1$. For PGD-2 and R-PGD-2, we let a larger perturbation size of $\nu = 2$ as recommended in the Adversarial ML Tutorial.

In the inner-loop, if the attack is iterative, we use a step-decay scheduler with initial step-size of 10, which decreases the step-size every 10 steps by a multiplicative factor of 0.2. In Table 1, we report the standard test accuracy as well as the adversarial test accuracy of the trained models over 10 independent random runs of the experiment. Different rows and columns correspond to different training algorithms and different attack models, respectively.

**Analysis.** We make the following observations in Table 1. First, reflecting the loss has a minimal effect on FGSM and BIM attacks, in terms of robust test accuracy of the trained models. In particular, the columns 1 and 2 (similarly columns 5 and 6) are identical up to the third decimal point.

Second, in PGD-2 attacks, reflecting the loss generally yields a stronger attack – note the striking differences in the last two columns between PGD-2 and R-PGD-2. We observe a milder trend for PGD-∞ attacks, where R-PGD-∞ attacks turns out to be only slightly stronger, except for the standard training setting where reflecting the loss has a huge impact on the robust error.

Third, we would like to remark on the performance of adversarially trained models. We can see that reflecting the loss in general helps robustness. In particular, second and fourth rows (PGD-∞ and PGD-2) are completely dominated by the third and fifth rows (R-PGD-∞ and R-PGD-2), respectively.

Finally, it is notable that even though PGD-2 and PGD-∞ are much weaker than their reflected counterparts, they are still competitive in terms of the robustness when used in adversarial training. Thissuggests that finding a "strong" attack is not a necessity for adversarial training to succeed.

### 4.3   Extension to multi-label setting

In binary classification using the logistic loss, in essence, adversarial training finds an attack that minimizes the log-likelihood of the correct class. Using the reflected loss, instead, we aim at maximizing the log-likelihood of the wrong class. In a multiclass classification scenario, there are multiple such wrong classes. Therefore, an important design question is which wrong class should be targeted in the attack phase? Here, we focus on the most natural choice: we target the wrong class with the highest log-likelihood. This greedy approach is easy to implement, and has minimal computational overhead over standard adversarial training.

We emphasize though that the greedy approach (described above) is sub-optimal, even in a simple linear setting. Intuitively, when the parameters are such that the logits for the true class correlate

with the logits for the most likely wrong class, the greedy approach fails. In particular, consider the following 3-class classification problem in $\mathbb{R}^2$. Let $f_W(x) = Wx$, where $W = [2e_1, e_1, 10e_2] \in \mathbb{R}^{3\times 2}$. Here, $e_i$ denotes the $i$-th standard basis. Consider the point $x = [1, 0]$. Clearly, class 1 and 3 have the highest and the smallest likelihoods, respectively. Given a perturbation size $\|x' - x\| \leq 0.3$, the likelihood of the second class will never dominate that of the first class:

$$w_1^\top(x + \delta) = 2e_1^\top(x + \delta) = 2(x_1 + \delta_1) > (x_1 + \delta_1) = e_1^\top(x + \delta) = w_2^\top(x + \delta),$$

where the inequality follows by using the fact that $x_1 = 1$ and $|\delta_1| \leq 0.3$. Therefore, the greedy approach fails here. Whereas, within the specified perturbation budget, maximizing the likelihood of the third class can indeed find a label-flipping attack. For example, with $\delta = [0, 0.3]$, the point $x' = [1, 0.3]$ will be assigned to the third class, because $w_3^\top x' = 3 > w_1^\top x = 2 > w_2^\top x = 1$.

| | Attack Size $\nu = 2/255$ | | | | | | | |
| --- | --- | --- | --- | --- | --- | --- | --- | --- |
| | Steps = 2 | | Steps = 4 | | Steps = 16 | | Steps = 32 | |
| | RA | SA | RA | SA | RA | SA | RA | SA |
| PGD | 14.182 | 91.254 | 20.702 | 90.424 | 21.014 | 90.132 | 20.848 | 90.09 |
| R-PGD | 14.338 | 91.208 | 20.726 | 90.384 | 20.958 | 90.06 | 20.746 | 89.992 |
| | Attack Size $\nu = 4/255$ | | | | | | | |
| PGD | 17.764 | 90.748 | 30.344 | 88.736 | 37.564 | 86.65 | 37.304 | 86.572 |
| R-PGD | 17.162 | 90.114 | 30.34 | 88.826 | 37.4 | 86.734 | 37.374 | 86.522 |
| | Attack Size $\nu = 8/255$ | | | | | | | |
| PGD | 20.064 | 90.478 | 34.21 | 87.746 | 48.916 | 78.402 | 48.936 | 77.926 |
| R-PGD | 20.1 | 90.564 | 34.19 | 87.852 | 48.792 | 78.382 | 48.828 | 77.982 |
| | Attack Size $\nu = 16/255$ | | | | | | | |
| PGD | 16.19 | 85.908 | 21.524 | 86.816 | 48.722 | 68.37 | 45.292 | 58.526 |
| R-PGD | 15.986 | 89.708 | 21.362 | 86.83 | 48.742 | 68.456 | 44.778 | 58.486 |

Table 2: Robust test accuracy (RA) of adversarially trained models with and without reflecting the loss, for different values of the attack size $\nu \in \{2, 4, 8, 16\}/255$ and number of steps in the attack Steps $\in \{2, 4, 16, 32\}$. We report the results for test-time attack size $\nu = 8/255$; the better performance is highlighted in gray, where the intensity corresponds to difference in performance.

We use adversarial training with and without reflected loss (denoted by R-PGD and PGD, respectively) to train a PreActResNet (PARN) He et al. [2016] on the CIFAR-10 dataset Krizhevsky et al. [2009]. In the training phase, we conduct experiments for attack size $\nu \in \{2, 4, 8, 16\}/255$. We build on the PyTorch implementation in Zhang et al. [2021], and we follow their experimental setup, which is described next. We use a SGD optimizer with a momentum parameter of $0.9$ and weight decay parameter of $5 \times 10^{-4}$. We set the batch size to 128 and train each model for 20 epochs. We use a cyclic scheduler which increases the learning rate linearly from 0 to $0.2$ within the first 10 epochs and then reduces it back to 0 in the remaining 10 epochs. We report robust test accuracy (RA) of an adversarially-trained model against PGD attacks Madry et al. [2018] (RA-PGD), where we take 50-step PGD with 10 restarts. We report the results for test-time attack size $\nu = 8/255$. Based on our empirical results, using the (greedy) reflected loss in adversarial training does not significantly impact the standard/robust generalization performance of the learned models.

## 5 Discussion

We study robust adversarial training of two-layer neural networks as a bi-level optimization problem. We propose *reflecting* the surrogate loss about the origin in the inner maximization phase when searching for an "optimal" perturbation vector to attack the current model. We give convergence guarantee for the inner-loop PGD attack and precise iteration complexity results for end-to-end adversarial training, which hold for any width and initialization under a margin assumption. We also provide an empirical study on the effect of reflecting the surrogate loss in real datasets. Next, we list few natural research directions for future work.

**Extension to multiclass setting.** In binary classification, which is the focus of this paper, reflecting the loss about the origin provides a concave lower-bound for the zero one loss (see Figure 1). Maximizing the reflected loss then corresponds to maximizing the likelihood of the wrong class. This simple modification enables us to guarantee the convergence of PGD-2 attacks, and yield stronger

attacks in our experiments. However, extending this idea to the multiclass setting is not trivial. In particular, the idea of maximizing the likelihood of the wrong class does not trivially generalize to the multiclass setting due to plurality of wrong classes. Nonetheless, as we show in the experimental section, a naive greedy approach to choose a wrong class seems to provide competitive performance in terms of standard/adversarial test error. Is there a simple, principled approach to obtain a lower-bound for the misclassification error in the multiclass setting? It would be interesting to explore theoretical and empirical aspects of such possible extensions.

**Beyond $\beta$-robustness.** The notion of $\beta$-robustness is crucial in our analysis. Although we provide robustness guarantees for arbitrarily small positive $\beta$ (see Corollary 3.6), our current analysis does not allow for standard robustness guarantees ($\beta = 0$) except for the linear setting ($\alpha = 1$). At a high level, the main challenge here is to guarantee that the attack can always find an adversarial example – if there exists one – regardless of whether the attack is $\beta$-effective or not. This is, in particular, challenging to establish for iterative attacks such as PGD, because they can only guarantee getting sufficiently close to an optimal attack in finite time. Therefore, if the optimal attack can just barely flip the sign, the computational time for finding it can grow unboundedly. Therefore, providing robust generalization guarantees ($\beta = 0$) is an interesting research direction for future work.

**Optimization geometry.** In our theoretical results, we focus on PGD-2 attacks, which are based on steepest descent with respect to the $\ell_2$ geometry. In our experiments, we also provide empirical results for steepest descent attacks with respect to $\ell_\infty$ geometry (including FGSM and BIM) on the reflected loss. We leave the theoretical analysis of such attacks to future work.

## Acknowledgements

This research was supported, in part, by DARPA GARD award HR00112020004, NSF CAREER award IIS-1943251, funding from the Institute for Assured Autonomy (IAA) at JHU, and the Spring'22 workshop on "Learning and Games" at the Simons Institute for the Theory of Computing.

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

Table 3: Key terms, corresponding symbols, and descriptions.

| Term | Symbol | Description |
|---|---|---|
| Net Function | $f(\mathrm{x}; \mathrm{W}), f_{\mathrm{W}}(x), f_{\mathrm{x}}(\mathrm{W})$ | $\sum_{r=1}^{m} a_r \sigma(\mathrm{w}_r \cdot \mathrm{x})$ |
| Leaky ReLU | $\sigma(z)$ | $\max\{\alpha z, z\}$ |
| Binary Xent | $\ell(z)$ | $\log(1 + e^{-z})$ |
| Mirrored Xent | $\ell_-(z)$ | $-\log(1 + e^z)$ |
| Robust Error | $\epsilon_{\mathrm{rob}}(\mathrm{W})$ | $\mathbb{P}[\max_{\delta \in \Delta(\mathrm{x})} y f_{\mathrm{W}}(\delta) < 0]$ |
| $\beta$-Robust Error | $\epsilon_\beta(\mathrm{W})$ | $\mathbb{P}[\max_{\delta \in \Delta(\mathrm{x})} y f_{\mathrm{W}}(\delta) < -\beta]$ |
| Net Grad wrt Weights | $\frac{\partial}{\partial \mathrm{w}_r} f_{\mathrm{x}}(\mathrm{W})$ | $a_r \sigma'(\mathrm{w}_r \cdot \mathrm{x})\mathrm{x}$ |
| Net Grad wrt Input | $\nabla_{\mathrm{x}} f_{\mathrm{W}}(\mathrm{x})$ | $\sum_{r=1}^{m} a_r \sigma'(\mathrm{w}_r \cdot \mathrm{x})\mathrm{w}_r$ |
| Homogeneity in Input | - | $f_{\mathrm{x}}(\mathrm{W}) = \langle \nabla_{\mathrm{x}} f_{\mathrm{W}}(\mathrm{x}), \mathrm{x} \rangle$ |
| Homogeneity in Weights | - | $f_{\mathrm{W}}(\mathrm{x}) = \langle \nabla_{\mathrm{W}} f_{\mathrm{x}}(\mathrm{W}), \mathrm{W} \rangle$ |

# A  Appendix

For simplicity of the notation, in our analysis of the PGD attack, we drop the subscripts denoting the iterates. That is, at the $t$-th iterate of the outer loop, the weight matrix and the sample point is denoted by W and x, $y$, instead of $\mathrm{W}_t$ and $\mathrm{x}_t, y_t$. We can then use the same variable $t$ to measure the progress of the attack in the inner loop. PGD updates are therefore given by $\Pi_{\Delta(\mathrm{x})}[\delta_t + \eta \nabla \ell_-(y f_{\mathrm{W}}(\delta_t))]$.

*Proof of Theorem 3.2.* We first bound the norm of the gradient of the iterates as follows:

$$
\begin{aligned}
\|\nabla f_{\mathrm{W}}(\delta_t)\| &= \|\sum_{r=1}^{m} a_r \sigma'(\mathrm{w}_r \cdot \delta_t)\mathrm{w}_r\| \\
&\leq \sum_{r=1}^{m} \|a_r \sigma'(\mathrm{w}_r \cdot \delta_t)\mathrm{w}_r\| \\
&\leq \kappa \sum_{r=1}^{m} \|\mathrm{w}_r\| \\
&\leq \kappa \sqrt{m} \|\mathrm{W}\|_F =: G
\end{aligned}
$$

We analyze the distance of iterates from $\delta_*$:

$$
\begin{aligned}
\|\delta_{t+1} - \delta_*\|^2 - \|\delta_t - \delta_*\|^2 &= \|\Pi[\delta_t + \eta_{\mathrm{att}} \nabla \ell_-(y f_{\mathrm{W}}(\delta_t))] - \delta_*\|^2 - \|\delta_t - \delta_*\|^2 \\
&\leq \|\delta_t + \eta_{\mathrm{att}} \nabla \ell_-(y f_{\mathrm{W}}(\delta_t)) - \delta_*\|^2 - \|\delta_t - \delta_*\|^2 \\
&= 2\eta_{\mathrm{att}} \langle \nabla \ell_-(y f_{\mathrm{W}}(\delta_t)), \delta_t - \delta_* \rangle + \eta_{\mathrm{att}}^2 \|\nabla \ell_-(y f_{\mathrm{W}}(\delta_t))\|^2 \\
&= 2\eta_{\mathrm{att}} \ell'_-(y f_{\mathrm{W}}(\delta_t)) y \langle \nabla f_{\mathrm{W}}(\delta_t), \delta_t - \delta_* \rangle + \eta_{\mathrm{att}}^2 \ell'_-(y f_{\mathrm{W}}(\delta_t))^2 \|\nabla f_{\mathrm{W}}(\delta_t)\|^2 \\
&\leq 2\eta_{\mathrm{att}} \ell'_-(y f_{\mathrm{W}}(\delta_t)) y \langle \nabla f_{\mathrm{W}}(\delta_t), \delta_t - \delta_* \rangle - \eta_{\mathrm{att}}^2 \ell_-(y f_{\mathrm{W}}(\delta_t)) G^2 \\
&\qquad\qquad (|\ell'_-(\cdot)| \leq \max\{1, -\ell_-(\cdot)\}) \\
&\leq 2\eta_{\mathrm{att}} \ell'_-(y f_{\mathrm{W}}(\delta_t)) (y f_{\mathrm{W}}(\delta_t) - y f_{\mathrm{W},t}(\delta_*)) - \eta_{\mathrm{att}} \ell_-(y f_{\mathrm{W}}(\delta_t)) \\
&\qquad\qquad (\eta_{\mathrm{att}} \leq 1/G^2) \\
&\leq 2\eta_{\mathrm{att}} (\ell_-(y f_{\mathrm{W}}(\delta_t)) - \ell_-(y f_{\mathrm{W},t}(\delta_*))) - \eta_{\mathrm{att}} \ell_-(y f_{\mathrm{W}}(\delta_t)) \\
&\qquad\qquad (\text{concavity}) \\
&\leq \eta_{\mathrm{att}} \ell_-(y f_{\mathrm{W}}(\delta_t)) - 2\eta_{\mathrm{att}} \ell_-(y f_{\mathrm{W},t}(\delta_*))
\end{aligned}
$$

where $f_{\mathrm{W},t}(\delta_*) := \langle \nabla f_{\mathrm{W}}(\delta_t), \delta_* \rangle$. Averaging over iterates, rearranging, and cancelling telescopic terms, we arrive at:

$$\frac{1}{T}\sum_{t=1}^{T} -\ell_-(yf_{\mathrm{W}}(\delta_t)) \leq \sum_{t=1}^{T} \frac{\|\delta_t - \delta_*\|^2 - \|\delta_{t+1} - \delta_*\|^2}{\eta_{\mathrm{att}}T} - \frac{2}{T}\sum_{t=1}^{T}\ell_-(yf_{\mathrm{W},t}(\delta_*))$$

$$\leq \frac{\|\delta_1 - \delta_*\|^2}{\eta_{\mathrm{att}}T} - 2\min_t \ell_-(yf_{\mathrm{W},t}(\delta_*)) \qquad \text{(Telescopic sum)}$$

$$\leq \frac{\nu^2}{\eta_{\mathrm{att}}T} - 2\min_t \ell_-(yf_{\mathrm{W},t}(\delta_*)) \qquad (\delta_1 = \mathrm{x}, \delta_* \in \Delta(\mathrm{x}))$$

$$\implies -\ell_-(yf_{\mathrm{W}}(\delta_{\mathrm{att}})) \leq \frac{\nu^2}{\eta_{\mathrm{att}}T} - 2\min_t \ell_-(yf_{\mathrm{W},t}(\delta_*))$$

Next, we show that if $\delta_*$ is an effective attack on $f_{\mathrm{W}}$, then it's also effective on $f_{\mathrm{W},t}$. In other words, we have that:

$$yf_{\mathrm{W},t}(\delta_*) = y\langle \nabla f_{\mathrm{W}}(\delta_t), \delta_* \rangle$$
$$= y\langle \nabla f_{\mathrm{W}}(\delta_*), \delta_* \rangle + y\langle \nabla f_{\mathrm{W}}(\delta_t) - \nabla f_{\mathrm{W}}(\delta_*), \delta_* \rangle$$
$$= yf_{\mathrm{W}}(\delta_*) + y\langle \nabla f_{\mathrm{W}}(\delta_t) - \nabla f_{\mathrm{W}}(\delta_*), \delta_* \rangle$$

Let $\mathcal{I}_t := \{j \in [m] | \sigma'(\mathrm{w}_j \cdot \delta_t) \neq \sigma'(\mathrm{w}_j \cdot \delta_*)\}$ be the set of nodes whose pre-activation sign at time $t$ is different from the optimal $\delta_*$. The second term can be bounded as follows:

$$y\langle \nabla f_{\mathrm{W}}(\delta_t) - \nabla f_{\mathrm{W}}(\delta_*), \delta_* \rangle = y\sum_{j=1}^{m} a_j \left(\sigma'(\mathrm{w}_j \cdot \delta_t) - \sigma'(\mathrm{w}_j \cdot \delta_*)\right)\mathrm{w}_j \cdot \delta_*$$

$$\leq |y|\sum_{j=1}^{m} |a_j \left(\sigma'(\mathrm{w}_j \cdot \delta_t) - \sigma'(\mathrm{w}_j \cdot \delta_*)\right)\mathrm{w}_j \cdot \delta_*|$$

$$= \kappa \sum_{j \notin \mathcal{I}_t} |\sigma'(\mathrm{w}_j \cdot \delta_t) - \sigma'(\mathrm{w}_j \cdot \delta_*)| \, |\mathrm{w}_j \cdot \delta_*|$$

$$\leq (1-\alpha)\kappa \sum_{j \notin \mathcal{I}} |\mathrm{w}_j \cdot \delta_* - \mathrm{w}_j \cdot \delta_t| \qquad (j \notin \mathcal{I}_t)$$

$$\leq (1-\alpha)\kappa \sum_{j \notin \mathcal{I}} \|\mathrm{w}_j\|\|\delta_* - \delta_t\| \qquad \text{(Cauchy-Schwarz)}$$

$$\leq (1-\alpha)\kappa\nu\sqrt{m}\|\mathrm{W}\|_F \qquad \text{(Cauchy-Schwarz)}$$

Given that there exist a $\beta$-effective, i.e. there exist $\delta_*$ such that $yf_{\mathrm{W}}(\delta_*) \leq -\beta\|\mathrm{W}\|_F$, we have:

$$yf_{\mathrm{W},t}(\delta_*) \leq yf_{\mathrm{W}}(\delta_*) + (1-\alpha)\kappa\nu\sqrt{m}\|\mathrm{W}\|_F$$

$$\leq yf_{\mathrm{W}}(\delta_*) + \frac{\beta}{2}\|\mathrm{W}\|_F \qquad (\nu \leq \frac{\beta}{2(1-\alpha)\kappa\sqrt{m}})$$

$$\leq \frac{1}{2}yf_{\mathrm{W}}(\delta_*).$$

Together with the previous inequality on the average of instantaneous loss, we arrive at:

$$-\ell_-(yf_{\mathrm{W}}(\delta_{\mathrm{att}})) = \min_{t \in [T]} -\ell_-(yf_{\mathrm{W}}(\delta_t))$$

$$\leq \frac{1}{T}\sum_{t=1}^{T} -\ell_-(yf_{\mathrm{W}}(\delta_t))$$

$$\leq \frac{\nu^2}{\eta_{\mathrm{att}}T} - 2\ell_-(yf_{\mathrm{W}}(\delta_*)/2).$$

Therefore, we have

$$2\ell_-(yf_{\mathrm{W}}(\delta_*/2)) \leq \ell_-(yf_{\mathrm{W}}(\delta_{\mathrm{att}})) + \frac{\nu^2}{\eta_{\mathrm{att}}T}. \tag{3}$$

Next, we show that for any $z, z'$, and $\epsilon > 0$, the inequality $2\ell_-(z/2) \leq \ell_-(z') + \epsilon$ implies that $-\ell'(z) \leq -2\ell'(z') + 4\epsilon$. The following inequalities hold true:

$$2\ell_-(z/2) \leq \ell_-(z') + \epsilon'$$

$$\implies -2\log(1 + e^{z/2}) \leq -\log(1 + e^{z'}) + \epsilon'$$

$$\implies 2\log(\frac{1}{1 + e^{z/2}}) \leq \log(\frac{e^{\epsilon'}}{1 + e^{z'}})$$

$$\implies \frac{1}{1 + e^z + 2e^{z/2}} \leq \frac{e^{\epsilon'}}{1 + e^{z'}}$$

$$\implies \frac{e^{-z}}{1 + e^{-z} + 2e^{-z/2}} \leq \frac{e^{-z'}}{1 + e^{-z'}} \cdot e^{\epsilon'}$$

$$\implies \frac{1}{2}\frac{e^{-z}}{1 + e^{-z}} \leq \frac{e^{-z'}}{1 + e^{-z'}} \cdot e^{\epsilon'} \qquad\qquad (2e^{-z/2} \leq 1 + e^{-z})$$

$$\implies -\ell'(z) \leq -\ell'(z') \cdot 2e^{\epsilon'} \qquad\qquad \text{(definition of } \ell'(\cdot))$$

$$\implies -\ell'(z) \leq -\ell'(z')2(1 + 2\epsilon') \qquad\qquad (e^z \leq 1 + 2z \text{ for all } z \in [0, 1])$$

$$\implies -\ell'(z) \leq -2\ell'(z') + 4\epsilon' \qquad\qquad (-\ell'(\cdot) \leq 1)$$

Let $z_* := yf_{\bar{W}}(\delta_*)$, $z_{\text{att}} := yf_{\bar{W}}(\delta_{\text{att}})$ and $\epsilon' := \frac{\nu^2}{\eta_{\text{att}}T}$. Using the above inequality, sub-optimality in terms of $\ell_-(\cdot)$, as given in Equation (3), implies sub-optimality in terms of $-\ell'(\cdot)$:

$$2\ell_-(z_*/2) \leq \ell_-(z_{\text{att}}) + \epsilon' \implies -\ell'(z_*) \leq -2\ell'(z_{\text{att}}) + 4\epsilon'$$

That is, for *any* $(x, y) \sim \mathcal{D}$, it holds with probability one that

$$-\ell'(yf_{\bar{W}}(\delta_*)) \leq -2\ell'(yf_{\bar{W}}(\delta_{\text{att}})) + \frac{4\nu^2}{\eta_{\text{att}}T}. \tag{4}$$

$\square$

Let $\mathbb{E}_{\text{AdvTr}}[\cdot]$ denote the expectation over a random draw of samples $(x_i, y_i)_{i=1}^t$ for Adversarial Training given in Algorithm 2. Let $\hat{H}_t := \langle W_t, V_* \rangle$, $\hat{G}_t^2 = \|W_t\|_F^2$, and let $H_t := \mathbb{E}_{\text{AdvTr}}[\hat{H}_t]$, $G_t^2 := \mathbb{E}_{\text{AdvTr}}[\hat{G}_t^2]$ be their corresponding population version.

*Proof of Lemma 3.7.* The proof heavily builds on the prior works of Frei et al. [2021]; we include it here for completeness. Let $V_* \in \mathbb{R}^{m \times d}$ be such that $v_r = \frac{1}{\sqrt{m}}\text{sgn}(a_r)v_*$. We define the correlation between the iterates and $V_*$ as follows $\hat{H}_t := \langle W_t, V_* \rangle$. This correlation evolves as:

$$\hat{H}_{t+1} = \langle W_{t+1}, V_* \rangle$$
$$= \langle W_t - \eta_{\text{tr}}\nabla\ell(y_t f_{\delta_t}(W_t)), V_* \rangle$$
$$= \hat{H}_t - \eta_{\text{tr}}\ell'(y_t f_{\delta_t}(W_t))y_t\langle \nabla_W f_{\delta_t}(W_t), V_* \rangle$$

Recall that $\frac{\partial}{\partial w_r}f_\delta(W) = a_r\sigma'(w_r \cdot \delta)\delta$. Therefore, we have that:

$$y_t\langle \nabla_W f_{\delta_t}(W_t), V_* \rangle = y_t \sum_{r=1}^m \langle a_r\sigma'(w_r \cdot \delta)\delta, \frac{1}{\sqrt{m}}\text{sgn}(a_r)v_* \rangle$$

$$= \frac{\kappa}{\sqrt{m}}\sum_{r=1}^m \sigma'(w_r \cdot \delta)y_t\langle \delta, v_* \rangle$$

Note that

$$y_t\langle \delta, v_* \rangle = y_t\langle x, v_* \rangle + y_t\langle \delta - x, v_* \rangle$$
$$\geq \gamma - |y_t\langle \delta - x, v_* \rangle|$$
$$\geq \gamma - \|\delta - x\|\|v_*\|$$
$$\geq \gamma - \nu$$

On the other hand, for leaky ReLU, it holds that $\sigma'(\cdot) \geq \alpha$. Therefore, we arrive at

$$y_t \langle \nabla_{\mathbf{W}} f_{\delta_t}(\mathbf{W}_t), \mathbf{V}_* \rangle \geq \frac{\kappa}{\sqrt{m}} \sum_{r=1}^{m} \alpha(\gamma - \nu)$$
$$= \kappa \alpha \sqrt{m}(\gamma - \nu)$$

Therefore, we have that

$$\hat{H}_{t+1} \geq \hat{H}_t - \eta_{\text{tr}} \ell'(y_t f_{\delta_t}(\mathbf{W}_t)) \alpha \kappa \sqrt{m}(\gamma - \nu)$$

$$\implies H_{T+1} \geq H_1 - \eta_{\text{tr}}(\gamma - \nu)\alpha \kappa \sqrt{m} \sum_{t=1}^{T} \mathbb{E}_{\texttt{AdvTr}} \ell'(y_t f_{\delta_t}(\mathbf{W}_t)) \quad \text{(taking expection } \mathbb{E}_{\texttt{AdvTr}}[\cdot])$$

The gradient norm is bounded as follows:

$$\|\nabla_{\mathbf{W}} f_{\delta_t}(\mathbf{W}_t)\|^2 = \sum_{j=1}^{m} \|a_j \sigma'(\mathbf{w}_{j,t} \cdot \delta_t)\delta_t\|^2 \leq m\kappa^2(R + \nu)^2$$

Next, we analyze the norm of the iterates, i.e., $\hat{G}_t^2 = \|\mathbf{W}_t\|_F^2$. It is also easy to verify that $-\ell'(z)z = \frac{ze^{-z}}{1+e^{-z}} = \frac{z}{1+e^z} \leq 1$. We have:

$$\hat{G}_{t+1}^2 = \|\mathbf{W}_t - \eta_{\text{tr}} \nabla \ell(y_t f_{\delta_t}(\mathbf{W}_t))\|^2$$
$$= \|\mathbf{W}_t\|_F^2 + \eta_{\text{tr}}^2 \|\nabla \ell(y_t f_{\delta_t}(\mathbf{W}_t))\|^2 - 2\eta_{\text{tr}} \ell'(y_t f_{\delta_t}(\mathbf{W}_t)) y_t \nabla_{\mathbf{W}} f_{\delta_t}(\mathbf{W}_t) \cdot \mathbf{W}_t$$
$$= \hat{G}_t^2 + \eta_{\text{tr}}^2 \ell'(y_t f_{\delta_t}(\mathbf{W}_t))^2 \|\nabla_{\mathbf{W}} f_{\delta_t}(\mathbf{W}_t)\|^2 - 2\eta_{\text{tr}} \ell'(y_t f_{\delta_t}(\mathbf{W}_t)) y_t f_{\delta_t}(\mathbf{W}_t)$$
$$\leq \hat{G}_t^2 + \eta_{\text{tr}}^2 m\kappa^2(R + \nu)^2 + 2\eta_{\text{tr}} \qquad (-\ell'(z)z \leq 1)$$
$$\leq \hat{G}_t^2 + 3\eta_{\text{tr}} \qquad (\eta_{\text{tr}} \leq m^{-1}\kappa^{-2}(R + \nu)^{-2})$$

Therefore, taking expectation $\mathbb{E}_{\texttt{AdvTr}}$ on both sides, we have that

$$G_{T+1}^2 \leq G_1^2 + 3\eta_{\text{tr}}T, \qquad (5)$$

and using $\sqrt{a + b} \leq \sqrt{a} + \sqrt{b}$, we have $G_T \leq G_1 + \sqrt{3\eta_{\text{tr}}T}$. Also, we have that $H_t^2 = (\mathbb{E}_{\texttt{AdvTr}} \langle \mathbf{W}_t, \mathbf{V}_* \rangle)^2 \leq \mathbb{E}_{\texttt{AdvTr}} \|\mathbf{W}_t\|_F^2 \|\mathbf{V}_*\|_F^2 \leq G_t^2$ so that $|H_t| \leq G_t$. Putting all together, we get:

$$-G_1 - \eta_{\text{tr}}(\gamma - \nu)\alpha \kappa \sqrt{m} \sum_{t=0}^{T-1} \mathbb{E}_{\texttt{AdvTr}} \ell'(y_t f_{\delta_t}(\mathbf{W}_t)) \leq H_0 - \eta_{\text{tr}}(\gamma - \nu)\alpha \kappa \sqrt{m} \sum_{t=0}^{T-1} \mathbb{E}_{\texttt{AdvTr}} \ell'(y_t f_{\delta_t}(\mathbf{W}_t))$$
$$\leq H_T$$
$$\leq G_T$$
$$\leq G_1 + \sqrt{3\eta_{\text{tr}}T}$$
$$-\eta_{\text{tr}}(\gamma - \nu)\alpha \kappa \sqrt{m} \sum_{t=0}^{T-1} \mathbb{E}_{\texttt{AdvTr}} \ell'(y_t f_{\delta_t}(\mathbf{W}_t)) \leq 2G_1 + 2\sqrt{\eta_{\text{tr}}T}$$

We now argue that for any $\epsilon$, there exist an iterate $t$ such that $-\mathbb{E}_{\texttt{AdvTr}} \ell'(y_t f_{\delta_t}(\mathbf{W}_t)) \leq \epsilon$. Assume otherwise, then we get:

$$\eta_{\text{tr}}(\gamma - \nu)\alpha \kappa \sqrt{m}\epsilon T \leq -\eta_{\text{tr}}(\gamma - \nu)\alpha \kappa \sqrt{m} \sum_{t=0}^{T-1} \mathbb{E}_{\texttt{AdvTr}} \ell'(y_t f_{\delta_t}(\mathbf{W}_t))$$
$$\leq 2G_1 + 2\sqrt{\eta_{\text{tr}}T}$$
$$\implies \eta_{\text{tr}}(\gamma - \nu)\alpha \kappa \sqrt{m}\epsilon \tau^2 - 2\sqrt{\eta_{\text{tr}}}\tau - 2G_1 \leq 0$$
$$\implies \tau \leq \frac{\sqrt{\eta_{\text{tr}}} + \sqrt{\eta_{\text{tr}} + 2G_1 \eta_{\text{tr}} \gamma \alpha \kappa \sqrt{m}\epsilon}}{\eta_{\text{tr}}(\gamma - \nu)\alpha \kappa \sqrt{m}\epsilon}$$
$$\implies T \leq \frac{4(1 + G_1 \gamma \alpha \kappa \sqrt{m}\epsilon)}{\eta_{\text{tr}}(\gamma - \nu)^2 \alpha^2 \kappa^2 m\epsilon^2}$$

Therefore, we have that $-\mathbb{E}_{\texttt{AdvTr}}\ell'(y_t f_{\delta_t}(W_t)) = -\mathbb{E}_{\texttt{AdvTr}}\ell'(y_t f_{W_t}(\delta_{\texttt{att}}(x_t))) \leq \epsilon$. Moreover,

$$\mathbb{E}_{\texttt{AdvTr}}[-\ell'(y_t f_{W_t}(\delta_{\texttt{att}}(x_t)))] = \mathbb{E}_{\mathcal{S}_t \sim \mathcal{D}^t}[-\ell'(y_t f_{W_t}(\delta_{\texttt{att}}(x_t)))]$$
$$\text{(independence from future samples)}$$
$$= \mathbb{E}_{\mathcal{S}_{t-1} \sim \mathcal{D}^{t-1}} \mathbb{E}_{(x_t, y_t) \sim \mathcal{D}} \left[ -\ell'(y_t f_{W_t}(\delta_{\texttt{att}}(x_t))) | \mathcal{S}_{t-1} \right]$$
$$\text{(Smoothing property of the conditional expectation)}$$
$$= \mathbb{E}_{\mathcal{S}} \mathbb{E}_{(x,y) \sim \mathcal{D}} [-\ell'(y f_{W_t}(\delta_{\texttt{att}}(x)))]$$
$$\text{($W_t$ independent of $(x_t, y_t)$ given $\mathcal{S}_{t-1}$)}$$

which completes the proof. $\qquad\qquad\square$

