# OpenReview forum: "Robustness Guarantees for Adversarially Trained Neural Networks"
_NeurIPS.cc/2023/Conference — NeurIPS 2023 poster_

### Official Review · Reviewer_8NN7 · 2023-06-16

**Soundness:** 3 good
**Presentation:** 3 good
**Contribution:** 3 good
**Rating:** 6
**Confidence:** 4

**Summary:**

This paper studies the optimization convergence of adversarial training in two-layer neural networks. This paper also proposes a reflecting loss which search for a better attack.

**Strengths:**

The paper is clear and easy to understand.

**Weaknesses:**

My major concern regarding this paper is that the condition on the attack strength $\nu$ is too strong. The constraint $\nu\leq \beta/(2(1-\alpha)\kappa\sqrt{m})$ is exactly used in Page 13 (in the appendix) so that the attack on the current model $f_{W,t}$ is closed to the one for the model $f_{W}$. However, in such a scenario, there is almost no difference between clean training and adversarial training. The authors need to relax this condition and provide more insights on the role of the attack strength and the difficulty when it gets larger.

Besides the major concern on the theoretical contribution, there are some issues in the experiments. First, the experiment in CIFAR-10 does not obtain a significant improvement. In Table 2, if counting the colored cells, PGD takes up 4, and R_PGD takes up 3, i.e., PGD may be even better than R-PGD in some cases. Second, there is no error bar information in this paper. Given that the difference between benchmarks and the proposed method is only of a small quantity, the authors need to run experiments for multiple times to obtain a standard error of the accuracies to verify that the improvements are statistically significant.

Finally, in addition to the weakness of the assumption on the attack strength, there is also another gap between the theoretical study and the experiments: the authors may consider theoretical justification on the generalization performance of adversarial training in the linear separable data. I would suggest the authors to investigate the effect on $\beta$ towards the generalization performance. In many existing literature, people study the generalization performance in simple models, e.g., Gaussian mixture model, linear regression model, but there is limited understanding on linear separable data.

**Questions:**

Please try to address my major concern (the first paragraph).

**Limitations:**

No ethical concern.

---

> ### Author Rebuttal · Authors · 2023-08-08
>
> 1.*Regarding attack strength*, the constraint $\nu \leq \beta/(2 (1-\alpha) \kappa \sqrt{m})$ is not too small since **$\kappa = 1/\sqrt{m}$; see line 134**.
>
> In fact, it is exactly what we should hope for. Consider, for example, the setting with $\alpha=0$ (i.e., linear unit). Then the constraint simplifies to $\nu \leq \beta/2$. If we think of $\beta$ as a hard margin between two classes and assume that the class conditional marginals are supported on hyperplanes that are a distance of $\beta$ from each other, then any perturbation of size larger than $\beta/2$ will result in robust test loss equal to one. We know that our results are tight since we recover several prior results when we set the parameters accordingly; see the remarks following Theorem 3.4.
>
>
>
> 2.*Regarding experiments*, the goal here is not to improve upon standard adversarial training. We know that adversarial training is quite successful in practice. This is what we state in the opening paragraph of the experimental results section (**see lines 232 — 235**). However, we cannot hope to give convergence guarantees for adversarial training as it involves maximizing a convex function in the inner loop which is not tractable. This is why we use a concave lower bound on 0-1 loss. **The question that remains is whether changing the adversarial training in this way does worse in practice. The answer is no.** It does not change the practice in any significant way but we gain new insights by establishing computational learning guarantees.
>
>
> 3.*Regarding linear separability*, we recall that the goal here is to give a bound on the computational complexity of adversarial training, i.e., how much time is needed to train a model that generalizes robustly. **Without linear separability, we know that empirical risk minimization is computationally hard even for simple models such as linear predictors.** So, if we cannot hope to say anything about the runtime of learning in a clean setting, there is little hope we can do so in the adversarial setting without assuming linear realizability. We know that it is a strong assumption, but unfortunately, there is not much to say in terms of runtime otherwise.

---

> > ### Comment · Reviewer_8NN7 · 2023-08-12
> >
> > I appreciate the authors' feedback.
> >
> > Regarding the attack strength, thanks for pointing out my misunderstanding. However, could you please provide some more intuitions on why the proof can work? From $\kappa=1/\sqrt{m}$, following paper
> >
> > Ba, J., Erdogdu, M., Suzuki, T., Wu, D., & Zhang, T. (2020, April). Generalization of two-layer neural networks: An asymptotic viewpoint. In International conference on learning representations.
> >
> > and
> >
> > Xing, Y., Song, Q., & Cheng, G. (2021, March). On the generalization properties of adversarial training. In International Conference on Artificial Intelligence and Statistics (pp. 505-513). PMLR.
> >
> > My understanding is that the neural network is expected to behave similar to a linear network, in which the attack simply follows the one in linear model. Or is there any difference in the intuition?

---

> > > ### Author Response · Authors · 2023-08-13
> > > **Responding to the follow-up comment from Reviewer 8NN7**
> > >
> > > Thank you for reading our rebuttal and for the follow-up questions. We will be sure to discuss these works in the revision.  Note though that there are significant differences between the settings/results of our work compared to those shared by the reviewer.
> > >
> > > The paper of Ba et al. focuses on the least squares regression problem with two-layer networks trained using gradient flow in a high dimensional setting wherein samples n, features d, and neurons h tend to infinity. We, on the other hand, focus on the problem of binary classification with two-layer networks trained using SGD. There is a fair bit of difference between the settings of the two papers. Furthermore, the paper of Ba et al. seeks to understand the double descent phenomenon and the role of inductive bias in different settings involving various initialization schemes, which is a very different focus from that of ours. We are not sure what connection we need to identify to be able to gain insights into why our proof works based on the work of Ba et al.
> > >
> > > The paper of Xing et al. focuses on statistical aspects of adversarial training for regression problems and makes some connection with the Lasso problem. They consider a lazy training setting and leverage the fact that the dynamics of training a two-layer neural network in such a setting is close to the dynamics of training a linear predictor. We on the other hand focus on binary classification, do not consider a lazy regime and focus on computational aspects of the learning problem (i.e., how much runtime is needed for robust learnability rather than the statistical complexity). There is very little connection between the settings and tools and methods used in the two works.
> > >
> > >
> > > Regarding the reviewer’s question about why the proof works. Our proof techniques actually do not leverage the lazy regime or somehow exploit the fact that the neural network behaves as a linear network. We make that amply clear in the paper in several places (e.g., see lines 39 — 42, lines 73-74, lines 154 — 156). The proof follows the convergence proof of the Perceptron algorithm after making appropriate changes to the nonlinear architecture of the neural networks. There are three key ideas: first, showing that the objective is non-convex and every critical point is a global minimum; second, showing that SGD converges to a global minimum after performing at most certain number of non-zero attributes. Finally, showing that as long as the attack size is smaller than the margin, robust training is not much harder than standard training. The reviewer may also want to check the following well-cited paper by Alon Brutzkus, Amir Globerson, Eran Malach, and Shai Shalev-Shwartz which laid the foundation for this work: https://arxiv.org/pdf/1710.10174.pdf.

---

> > > > ### Comment · Reviewer_8NN7 · 2023-08-13
> > > >
> > > > Thanks for the response. I now better understand the contribution of this paper, and I have increased my score from 4 to 6.

---

> > ### Comment · Area_Chair_rvPi · 2023-08-12
> >
> > Hi Reviewer 8NN7,
> >
> > Please let me know whether the response addressed your concern.
> >
> > Best
> >
> > AC

---

> > > ### Comment · Reviewer_8NN7 · 2023-08-12
> > >
> > > Hi AC,
> > >
> > > Thanks for your message. I may need more information on the novelty of the proof technique.
> > >
> > > Regards,

---

### Official Review · Reviewer_Gmw4 · 2023-06-29

**Soundness:** 3 good
**Presentation:** 3 good
**Contribution:** 3 good
**Rating:** 7
**Confidence:** 3

**Summary:**

The paper studies robust training in 2-layer networks. This problem is studied as a 2-spet procedure - finding adversarial samples, and training over these samples. The main result is given a linearly separable dataset, then a 2-layer network with leaky-ReLU activation trained robustly with SGD converges to a robust network in polynomial time. Several experiments are given to support the theoretical results.

**Strengths:**

- I think it is interesting to study robust optimization in a rigorous way, and the main result of this paper- namely that robust SGD converges to a robust network in polynomial time is nice.
- The empirical result complements the theoretical results, while also generalizing to a multi-class setting.
- I think the proof sketch gives a nice and intuitive explanation of the main result


**Weaknesses:**

- The use of the reflected loss seems like a crucial part of the proof, although I don’t think it is motivated enough. Is it used just for some technical part of the proof? or is there more to it? I think if it is indeed a crucial part, then there should be a more in-depth explanation of why it is used.
- The model analyzed is a 2-layer network without biases.  It is not clear whether removing the bias is done just for simplicity, or it is indeed necessary. I think the proof uses homogeneity w.r.t x, which may be the cause for the bias-less network, but if so, can the results be generalized to networks with bias terms?
- More of a suggestion: Are the convergence results of algorithms 1 & 2 can be decoupled? i.e. will Theorem 3.4 about the convergence of algorithm 2 will hold when instead of algorithm 1 we use any other attack? If so, I think it is good to mention it.


**Questions:**

- Will the proof work also on networks with bias terms?
- Why is the reflection loss used in the analysis?


**Limitations:**

The authors adequately address the limitations of their work.

---

> ### Author Rebuttal · Authors · 2023-08-09
>
> [Q1] We think that the proof should go through even with bias terms as well, but it may require some extra work. In particular, some of the recent work (e.g., see https://arxiv.org/pdf/2102.11840.pdf and https://arxiv.org/pdf/2301.00327.pdf). The idea here is similar to how the proof goes through in the NTK setting — i.e., ensuring that the weights and the bias terms do not move too far from the initialization, which is indeed the case in many settings as these works show.
>
> [Q2] The reason that the reflected loss is used in the analysis is that it is a concave function that is a lower bound on the 0-1 loss. See, the technical challenge is that the inner loop of adversarial training for finding an adversarial attack amounts to maximizing a convex function which is computationally intractable. It is also unclear why one would use an upper bound on the 0-1 loss function if one were trying to maximize it. So, instead, we use a concave lower bound which is intuitive and computationally tractable. We will add a discussion to that effect.
>
> Regarding your suggestion of decoupling the results of the two algorithms, that is indeed the case. That is actually how we structured Section 3. Theorem 3.2 is for Algorithm 1 and Theorem 3.4 is for Algorithm 2 which involves Algorithm 1. It is also possible to use a different procedure other than PGD — as long as that algorithm returns a good enough attack vector in polynomial runtime, we can still give analogs of Theorem 3.4. We will think more about other alternatives. Thanks for your suggestion.

---

> > ### Comment · Reviewer_Gmw4 · 2023-08-13
> > **Re: Rebuttal**
> >
> > I thank the authors for the response.

---

### Official Review · Reviewer_2oBp · 2023-07-02

**Soundness:** 3 good
**Presentation:** 3 good
**Contribution:** 3 good
**Rating:** 6
**Confidence:** 4

**Summary:**

This paper studies the convergence of adversarial training of two layer neural network, with gradient descent ascent (GDA) type algorithm. The algorithm considered here solvies inner max problem on a surrogate concave loss, and solves outer minimization problem on log-exp loss. Finally, the authors show the convergence to $\epsilon$ weaker robust error with no more than $O(1/\epsilon^2)$ iterations. The proof contains two parts, first they show that, solving the surrogate concave loss will yield a pertubation, which is almost as good as the true optimal adversarial perturbation, and then they borrow the existing analysis of two layer classification neural network training, to show that under the solved perturbation, the outer minimization problem can converge to global minima. To the best of my knowledge, this is the first work towards analyzing behavior of GDA on adversarial neural network training.

**Strengths:**

 The convergence of adversarial neural network training is a longstanding problem. The most relevant work is [Gao et al 2019], where they assume the inner max problem is solved by some oracle, not PGD, and hence they only need to perform minimization convergence analysis on the adversarial loss. This paper, for the first time, consider algorithm-based convergence, where they study the case that the inner max problem is solved by projected gradient ascent.



**Weaknesses:**

1. My main concern is that, the proof seems to be heavily depending on previous result, and the main technical novelty is the idea of running PGD on concave surrogate, and show the converegnce of PGD.

2. Running PGD on concave surrogate loss is not widely used in practice. In practice, the gradient descent and ascent steps are usually performed on the same loss function.

**Questions:**

1. I notice that [Allen-Zhu and Li 2022] also studies adversarial training, where they consider Fast gradient method to solve inner max problem. Could the authors do some comparison with this paper?



Allen-Zhu, Zeyuan, and Yuanzhi Li. "Feature purification: How adversarial training performs robust deep learning." In 2021 IEEE 62nd Annual Symposium on Foundations of Computer Science (FOCS), pp. 977-988. IEEE, 2022.

**Limitations:**

One limitation as I mentioned is that the algorithm used here is not exactly consistent with what people use in practice. I do not see any societal impact.

---

> ### Author Rebuttal · Authors · 2023-08-08
>
> [W1] Regarding the “*main technical novelty is the idea of running PGD on concave surrogate, and show the converegnce of PGD*”. That is a fair characterization of the main contribution. The computational learning guarantee for the end-to-end adversarial training follows as a corollary once we have the result for the PGD attack. Note though that the result for the PGD attack is novel and it required us to rethink adversarial training. While, in hindsight, things look easy once you know the trick, it was not quite straightforward. We also believe that simple ideas are elegant and often powerful.
>
> [W2] Regarding the comment about “*Running PGD on concave surrogate loss is not widely used in practice. In practice, the gradient descent and ascent steps are usually performed on the same loss function.*” That is true. However, we can never hope to analyze that approach since maximizing a convex function is computationally intractable. What we do show is that running PGD on concave surrogate is principled and does not change the practice by much. This is evidenced by our empirical results. Both approaches are actually quite comparable.
>
> [Q1] Yes, we would be happy to add a discussion of how our approach and results compare with that of Allen-Zhu and Li. Note though that the two settings are somewhat incomparable as Allen-Zhu and Li make various distributional assumptions, e.g., the sparse coding model. Nonetheless, we do agree that some comparisons on how the two works handle the inner loop maximization are in order.

---

> > ### Comment · Reviewer_2oBp · 2023-08-12
> > **Thanks for your rebuttal**
> >
> > Thanks for providing explanation to my concerns. Now I see why the authors consider to maximize the surrogate loss since maximizing convex loss is not feasible to analyze. I change my score to 6 since this is the first work analyzing the GDA type of algorithm on neural network training, and I recommend to accept. I also suggest the authors can discuss more relevant works like Allen-zhu and Li's FOCS paper, on the difference of handling the inner maximization problem.

---

### Official Review · Reviewer_avt7 · 2023-07-09

**Soundness:** 2 fair
**Presentation:** 3 good
**Contribution:** 3 good
**Rating:** 6
**Confidence:** 4

**Summary:**

This paper investigates the adversarial training of two-layer neural networks on linearly separable data. The authors propose to reflect the commonly used convex surrogate loss during the inner loop that generates adversarial attack via the PGD method, and derives guarantee on the convergence of the attack. Meanwhile, this paper also provides theoretical results on iteration complexity for the adversarial training on linearly separable data that holds for any width and initialization of the network. Numerical studies are conducted to show the performance.

**Strengths:**

1.	The investigated topic of adversarial training of neural networks is important and has many applications in real world, while the theoretical aspects have not been well-studied. The results presented in this paper are non-trivial contribution to the field, and might help with better understanding of adversarial training of neural networks in more complicated settings.
2.	The theoretical analysis seems solid. The authors introduce some interesting terms such as $\beta$-effective attack and $\beta$-robust to help the analysis, which might help in other study.
3.	Overall, the paper is well-organized and easy to follow.


**Weaknesses:**

1.	The empirical results with MNIST and CIFAR-10 do not show significant difference between the performance of standard adversarial training and adversarial training using proposed reflected loss function.

2.	There are some space of improvement regarding the experiments. Specifically, please consider the followings:

(1)	The result presented in Figure 2 does not seem convincing to me because (a) the setting is too simple and hand-crafted, while the real data can behave very differently; (b) Only one specific data point ($x=[3,2,1]$) is considered. It is doubtful whether it is cherry-picked or not.

(2)	In Table 1, the robust testing error for standard training model under standard PGD attack (0.033) is much smaller than FGSM (0.286), which is unlikely since PGD usually generates stronger attack through multiple iterations. Please double-check the correctness.

(3)	In Table 1, please consider including the evaluation of different models over clean testing data (un-attacked) for better comparison. Also, for Tables 1 and 2, please consider including the evaluation under other attacks, e.g., CW attack [1] and AutoAttack [2].

Please also consider my questions in the next part.


[1] Carlini, N., & Wagner, D. (2017, May). Towards evaluating the robustness of neural networks. In 2017 ieee symposium on security and privacy (sp) (pp. 39-57). ieee.

[2] Croce, F., & Hein, M. (2020, November). Reliable evaluation of adversarial robustness with an ensemble of diverse parameter-free attacks. In International conference on machine learning (pp. 2206-2216). PMLR.


**Questions:**

1.	What feature of the reflected loss function enables the derivation of the theoretical results, or why the standard loss function make the analysis difficult? Can one replace the reflected loss function with a more general loss function?

2.	Theorem 3.4 says that "in at most $T_{tr}\cdots$ iterations, Algorithm 2 $\cdots$ $\textit{finds an iterate}$ $\tau$” with certain property. What does that mean exactly? Is it guaranteed that the model at the end of iteration $T_{tr}$ maintain such property? Or, is any early-stopping criteria necessary in order to guarantee the property?


**Limitations:**

The authors have adequately addressed the limitations of the work.

---

> ### Author Rebuttal · Authors · 2023-08-08
>
> [W1] Regarding “*The empirical results with MNIST and CIFAR-10 do not show significant difference between the performance of standard adversarial training and adversarial training using proposed reflected loss function.*” That is actually what we were hoping for.
>
> The goal here is not to improve upon standard adversarial training. We know that adversarial training is quite successful in practice. This is what we state in the opening paragraph of the experimental results section (**see lines 232 — 235**). However, we cannot hope to give convergence guarantees for adversarial training as it involves maximizing a convex function in the inner loop which is not tractable. This is why we use a concave lower bound on 0-1 loss. **The question that remains is whether changing the adversarial training in this way does worse in practice. The answer is no.** It does not change the practice in any significant way but we gain new insights by establishing computational learning guarantees.
>
> [W2] (1) Regarding “*improving experiments*”, we note that the point of experiment in Figure 2 was to help compare the utility of surrogate loss vs the reflected loss at finding the attack vectors. **It is meant to be a toy example so that we can carefully understand what is going on rather than simply report numbers that often are not very insightful. The goal is here not to have a leaderboard competition, but to unravel the inner workings of adversarial training.**
>
> [W2] (2) It is perhaps not very informative to compare performance across different attacks since the noise budget is not quite “calibrated.” In other words, it is unclear what the noise perturbation budget for an $\ell_2$ attack be for a certain noise budget corresponding to $\ell_\infty$ attack or the FGSM attack.
>
> [W2] (3) Sure, we will be happy to do so. If you look at the code we provide, it is actually quite straightforward to run the corresponding experiments for other attacks and respective reflected losses.
>
> [Q1] The feature of reflected loss that allows our analysis is that it is a concave function that is a lower bound on the 0-1 loss. See, the technical challenge is that the inner loop of adversarial training for finding an adversarial attack amounts to maximizing a convex function which is computationally intractable. It is also unclear why one would use an upper bound on the 0-1 loss function if you were trying to maximize it. So, instead, we use a concave lower bound which is intuitive and computationally tractable. That is the key feature.
>
> [Q2] That is a poor way of writing. We should have written “after $T_\textrm{tr}$ iterations”. In any case, what we mean is that after $T_\textrm{tr}$ iterations we find a model that is guaranteed to have a small robust error.

---

> > ### Comment · Reviewer_avt7 · 2023-08-11
> > **Follow-up comment on W2-2**
> >
> > Thank the authors for clarification during the discussion period. Most of my concerns get addressed. Here is my follow-up comment regarding W2-2.
> >
> > I think the significant difference between the robust testing error for standard training model under standard PGD attack vs FGSM attack (0.286) is not due to the perturbation budget. Note that the reported robust testing error for PGD-$\infty$ (0.033) and PGD-2 (0.003) are both much smaller than 0.286 as reported for FGSM attack. Considering that PGD takes multiple iterations compared to FGSM and is more powerful in searching a strong attack, I still do not think this difference is reasonable. Did you repeat this experiment for a few times?

---

> > > ### Author Response · Authors · 2023-08-13
> > > **Regarding Follow-up comment from Reviewer avt7**
> > >
> > > Yes, we did repeat the experiments multiple times and we do not believe there is a problem with the experiments. We provide the code as part of the supplement and it is fairly simple to check and run the code.
> > >
> > > We do note that the size of $\nu$ is an order of magnitude different for FGSM vs. PGD. Given our simple setting, the difference here could have been effectively even larger. Please refer to lines 267 -- 269 where we write:
> > >
> > > "The perturbation size for FGSM, PGD-$\infty$, and BIM (and their corresponding reflected version) is set to $\nu = 0.1$. For PGD-2 and R-PGD-2, we let a larger perturbation size of $\nu = 2$ as recommended in the Adversarial ML Tutorial."

---

> > > > ### Comment · Reviewer_avt7 · 2023-08-21
> > > >
> > > > Thank the authors for all the response again. I have changed my rating from 5 to 6, considering the theoretical contribution of the work.

---

### Comment · Area_Chair_rvPi · 2023-08-11

Hi all,

Thanks for serving as the reviewers for this submission. As the authors have already turned in their responses. It is our turn to start the further discussion. Here is a to-do list:

(1) Please acknowledge the authors when you finish reading their responses.
(2) Please indicate whether you have any further questions for the authors such that they can continue to response.
(3) Please indicate whether you are willing to change the ratings.

Best

AC

---

### Decision · Program_Chairs · 2023-09-21

**Decision:**

Accept (poster)

**Comment:**

This paper provides a theoretical analysis of robust adversarial training of neural networks using projected gradient descent for the inner maximization problem. The theoretical results demonstrating polynomial time convergence of this approach are novel and represent an important contribution. The reviewers found the paper to be technically solid, well-written, and easy to follow. The empirical results were viewed as complementary and helped validate the theory. Overall, the reviewers were positive about the paper and felt it provides new theoretical insights into an important problem. The main unaddressed concern was to better differentiate the approach from closely related work by Allen-Zhu and Li. However, the reviewers felt the paper makes a significant enough contribution as is. Considering the reviewers' overwhelmingly positive feedback, I recommend accepting this paper. The authors have advanced the theoretical understanding of adversarial training convergence and provided results that can inform future work in this area.